# Novel Ratio Soluble Fms-like Tyrosine Kinase-1/Angiotensin-II (sFlt-1/ANG-II) in Pregnant Women Is Associated with Critical Illness in COVID-19

**DOI:** 10.3390/v13101906

**Published:** 2021-09-23

**Authors:** Salvador Espino-y-Sosa, Raigam Jafet Martinez-Portilla, Johnatan Torres-Torres, Juan Mario Solis-Paredes, Guadalupe Estrada-Gutierrez, Jose Antonio Hernandez-Pacheco, Aurora Espejel-Nuñez, Paloma Mateu-Rogell, Angeles Juarez-Reyes, Francisco Eduardo Lopez-Ceh, Jose Rafael Villafan-Bernal, Lourdes Rojas-Zepeda, Iris Paola Guzman-Guzman, Liona C. Poon

**Affiliations:** 1Clinical Research Deparment, Instituto Nacional de Perinatologia Isidro Espinosa de los Reyes, Mexico City 11000, Mexico; salvadorespino@gmail.com (S.E.-y.-S.); raifet@hotmail.com (R.J.M.-P.); juan.mario.sp@gmail.com (J.M.S.-P.); gpestrad@gmail.com (G.E.-G.); dr.antoniohernandezp@gmail.com (J.A.H.-P.); auro.espejel@gmail.com (A.E.-N.); dramateurogell@gmail.com (P.M.-R.); 2Iberoamerican Research Network in Obstetrics, Gynecology and Translational Medicine, Mexico City 06720, Mexico; joravibe@gmail.com; 3Maternal Fetal Medicine Department, Hospital General de Mexico, “Dr. Eduardo Liceaga”, Mexico City 06720, Mexico; dra.angejuarez@gmail.com (A.J.-R.); doctorceh.mx@gmail.com (F.E.L.-C.); 4Laboratory of Immunogenomics and Metabolic Diseases, Instituto Nacional de Medicina Genomica, Mexico City 14610, Mexico; 5Maternal Fetal Medicine Department, Instituto Materno Infantil del Estado de Mexico, Mexico City 50170, Mexico; dra.rojaszepeda@gmail.com; 6Faculty of Chemical-Biological Sciences, Autonomous University of Guerrero, Chilpancingo 39086, Mexico; ipguzman2@gmail.com; 7Department of Obstetrics and Gynaecology, The Chinese University of Hong Kong, Hong Kong, China; liona.poon@cuhk.edu.hk

**Keywords:** COVID-19, maternal death, angiotensin-II, sFlt-1

## Abstract

Background: In healthy pregnancies, components of the Renin-Angiotensin system (RAS) are present in the placental villi and contribute to invasion, migration, and angiogenesis. At the same time, soluble fms-like tyrosine kinase 1 (sFlt-1) production is induced after binding of ANG-II to its receptor (AT-1R) in response to hypoxia. As RAS plays an essential role in the pathogenesis of COVID-19, we hypothesized that angiogenic marker (sFlt-1) and RAS components (ANG-II and ACE-2) may be related to adverse outcomes in pregnant women with COVID-19; Methods: Prospective cohort study. Primary outcome was severe pneumonia. Secondary outcomes were ICU admission, intubation, sepsis, and death. Spearman’s Rho test was used to analyze the correlation between sFlt-1 and ANG-II levels. The sFlt-1/ANG-II ratio was determined and the association with each adverse outcome was explored by logistic regression analysis and the prediction was assessed using receiver-operating-curve (ROC); Results: Among 80 pregnant women with COVID-19, the sFlt-1/ANG-II ratio was associated with an increased probability of severe pneumonia (odds ratio [OR]: 1.31; *p* = 0.003), ICU admission (OR: 1.05; *p* = 0.007); intubation (OR: 1.09; *p* = 0.008); sepsis (OR: 1.04; *p* = 0.008); and death (OR: 1.04; *p* = 0.018); Conclusion: sFlt-1/ANG-II ratio is a good predictor of adverse events such as pneumonia, ICU admission, intubation, sepsis, and death in pregnant women with COVID-19.

## 1. Introduction

SARS-CoV-2 infection and its symptomatic disease (COVID-19) are nowadays one of the leading causes of death worldwide. Different studies have shown that pregnant women with COVID-19 are at increased risk of serious illness, such as pneumonia (relative risk [RR]: 1.97; 95% CI: 1.82–2.13) and death (RR: 1.68; 95% CI: 1.36–2.08), than matched reproductive-age non-pregnant women [1,2].

The Renin-Angiotensin system (RAS) is now known to play an essential role in the pathogenesis of COVID-19 [3]. Typically, renin cleaves angiotensinogen into Angiotensin-I (ANG-l), ANG-l (physiologically inactive) is converted into Angiotensin-II (ANG-II) by the Angiotensin-Converting Enzyme-1 (ACE-1), and ANG-II is transformed into ANG 1–7 by Angiotensin-Converting Enzyme-2 (ACE-2), regulating the cardiovascular and renal function [4]. In target tissues (alveolar epithelial cells, intestinal epithelial cells, and endothelial cells), SARS-CoV-2 spike (S) protein binds to ACE-2, causing a reduction of the ACE-2 receptor in the membrane, potentially impairing ANG-II balance [5,6].

In healthy pregnancies, RAS components are present in the placental villi and contribute to placental invasion, migration, and angiogenesis [7]. Furthermore, RAS components help promote placental circulation and blood flow, facilitating fetal oxygenation [8]. Expression of ACE-2 and TMPRSS2 decreases in late gestation [9]. This downregulation of RAS is associated with adverse maternal-perinatal outcomes, particularly pre-eclampsia and fetal growth restriction [10]. To date, there are no original studies in pregnant women with COVID-19 reporting blood concentrations of ACE-2 and ANG-II, two molecules that are potentially involved in the pathogenesis of severe disease in COVID-19 [11,12].

On the other hand, soluble fms-like tyrosine kinase 1 (sFlt-1) is a protein related to placental hypoxia, endothelial damage, sepsis, and acute lung injury (15) [13]. sFlt-1 production is induced when ANG-II binds to its receptor (AT1) as a response to hypoxia [14]. It is known that sFlt-1 causes endothelial dysfunction, sensitizing the endothelial cells to the effect of ANG-II in the whole endothelium and placenta [15]. In critically ill non-pregnant COVID-19 patients, there is an upregulation of sFlt-1, suggesting that this protein might play a role in the COVID-19-associated systemic endothelial dysfunction [16,17].

We hypothesized that the primary endothelial dysfunction component characterized by an imbalance between sFlt-1 as an angiogenic marker and RAS components (ANG-II and ACE-2) may be related to adverse outcomes in pregnant women with COVID-19.

Therefore, this study aims to investigate the association between serum concentrations of sFlt-1 and RAS components to adverse outcomes in pregnant women with COVID-19.

## 2. Materials and Methods

### 2.1. Study Design and Participants

We conducted a prospective cohort study at the National Institute of Perinatology “Isidro Espinosa de los Reyes” and the General Hospital of Mexico “Dr. Eduardo Liceaga”, both third reference hospitals in Mexico City. Inclusion criteria were all pregnant women who arrived at the emergency department with respiratory symptoms and a positive RTqPCR for SARS-CoV-2 between December 2020 and July 2021. The study protocol was prospectively approved by the Ethics and Research Committee of the National Institute of Perinatology (2020–1-32). All enrolled women provided written informed consent.

### 2.2. Data Collection

The following data were collected from the medical records: age, gestational age, pregestational body mass index (pBMI [kg/m^2^]), chronic hypertension, pre-gestational diabetes, mean arterial pressure (MAP), pneumonia, sepsis, acute renal failure, organ dysfunction, ICU admission, intubation, and mortality. The following pregnancy outcomes were recorded: preeclampsia (defined according to The American College of Obstetricians and Gynecologists) [18], preterm birth (birth < 37 weeks’ gestation), birth weight, Apgar score at the 1st and 5th min, neonatal asphyxia, respiratory distress syndrome (RDS), neonatal sepsis, the requirement of neonatal intensive care unit (NICU) admission, and neonatal death. Blood samples were obtained at hospital admission, and the following laboratory results were recorded: leukocytes, neutrophils, lymphocytes, hemoglobin, hematocrit, platelets, glucose, creatinine, uric acid, aspartate aminotransferase (AST), alanine aminotransferase (ALT), direct bilirubin, indirect bilirubin, triglycerides, cholesterol, D-dimer, fibrinogen, partial thromboplastin time (PTT), prothrombin time (PT), C-reactive protein (C-RP), and procalcitonin. These parameters are routinely tested in pregnant women with COVID-19.

### 2.3. Plasma Measurements of ACE-2, ANG-II, and sFlt-1

Upon admission, an additional blood sample was obtained specifically for research purposes. The blood sample was centrifuged for 10 min at 1000× *g*. Plasma was separated, aliquoted, and stored at −70 °C until analysis. ELISA commercial kits were used to measure ACE-2 (Aviscera Bioscience, Santa Clara, CA. USA. cat SK00707-01) and ANG-II (Enzo Life Sciences, Farmingdale, NY. USA. cat ADI-900-204) according to the manufacturer’s instructions and analyzed in a Synergy HT plate reader (BioTek, Winooski, VT, USA). PlGF (Elecsys PlGF, Roche^®^) and sFlt-1 (Elecsys sFlt-1, Roche^®^) levels were measured by electrochemiluminescence using an automated analyzer cobas-e411 (Roche Diagnostics^®^, CH) according to the manufacturer’s instructions.

### 2.4. Outcome

The primary outcome was pregnant women with severe pneumonia. Secondary outcomes were ICU admission, intubation, viral sepsis, and maternal death as a direct result of SARS-CoV-2 infection. Severe pneumonia was defined according to the American Thoracic Society criteria, which include either one major criterion (septic shock with need for vasopressors, or respiratory failure requiring mechanical ventilation) or three or more minor criteria (respiratory rate ≥ 30 breaths/min; PaO_2_/FIO_2_ ratio ≤ 250; multilobar infiltrates; confusion/disorientation; uremia [blood urea nitrogen level ≥ 20 mg/dL]; leukopenia [white blood cell count < 4000 cells/µL]; thrombocytopenia [platelet count < 100,000/ µL]; hypothermia [core temperature < 36 °C]; hypotension requiring aggressive fluid resuscitation) [19,20]. ICU admission was decided according to the Quick Sequential Organ Failure Assessment (qSOFA) score, where a score of ≥2 points would require ICU admission [21]. Viral sepsis was defined according to the Sepsis-3 International Consensus [22] associated with SARS-CoV-2 infection [23].

### 2.5. Statistical Analysis

Descriptive and inferential statistics were used. Quantitative variables were reported as the median and interquartile range (IQR), while qualitative data were reported as numbers and percentages. Differences between variables among COVID-19-severity were compared using the Mann–Whitney U test or X^2^ test. We performed a correlation between all biochemical parameters to explore possible candidates for multiple logistic regression. All significant candidates were explored in an adjusted logistic regression to establish independent predictors for adverse outcomes. To explore a possible endothelial dysfunction, we performed sFlt-1/PlGF and sFlt-1/ANG-II ratios. Forward and backward stepwise logistic regression analyses were performed to assess the association between independent variables and primary and secondary outcomes including all possible candidates in the correlation analysis. The adjusted model’s performance after logistic regression was evaluated by receiver-operating-curve (ROC) analysis to estimate the area under the curve. *p*-values < 0.05 were considered statistically significant. (StataCorp. 2020. Stata Statistical Software: Release 17. College Station, TX, USA: StataCorp LLC.).

## 3. Results

### 3.1. Description of the Cohort and Characteristics of the Study Population

A total of 80 pregnant women with SARS-CoV-2 infection were included for the analysis. Twenty-five (31.25%) had severe COVID-19 disease and 55 were classified as non-severe. There were two (2.5%) maternal deaths. Baseline characteristics were similar between groups (Table 1).

There were no significant differences in the rate of preeclampsia between severe and non-severe COVID-19. Among the 25 severe cases, 24 (96%) were delivered by Cesarean section, 6 (24%) had neonatal asphyxia, 10 (40%) had RDS, and 11 (44%) required NICU admission, including 7 (28%) neonatal deaths (Appendix A).

Women with severe pneumonia had higher levels of AST, direct bilirubin, C-RP, sFlt-1, procalcitonin, sFlt-1/PlGF ratio, and sFlt-1/ ANG-II ratio. The severe group had lower levels of lymphocytes, total cholesterol, and ANG-II (Table 2).

### 3.2. Correlation between sFlt-1 and ANG-II

Spearman´s Rho test was used to identify the relationship of sFlt-1 and ANG-II, and a significant correlation was found among severe pneumonia (r = −0.453; *p* < 0.001) (Figure 1).

### 3.3. Association with the Primary and Secondary Outcomes

There was a significant association between severe pneumonia in women with SARS-CoV-2 infection and sFlt-1/ANG-II ratio (OR: 1.31; 95% CI: 1.09–1.56; *p* = 0.003) (Table 3). Among secondary outcomes, sFlt-1/ANG-II ratio was associated with ICU admission (OR: 1.05; 95% CI: 1.01–1.09; *p* = 0.007); intubation (OR: 1.09; 95% CI: 1.02–1.16; *p* = 0.008); viral sepsis (OR: 1.04; 95% CI: 1.01–1.08; *p* = 0.008); and maternal death (OR: 1.04; 95% CI: 1.00–1.07; *p* = 0.018) (Appendix A).

### 3.4. sFlt-1/ANG-II Ratio for the Prediction of Adverse Outcomes in COVID-19

The AUC of sFlt-1/ANG-II ratio for the prediction of severe pneumonia by COVID-19 was 0.9608 (95% CI: 0.807–0.981). The detection rates for severe pneumonia at 5% and 10% false-positive-rate were 52% and 88%, respectively (Figure 2). The best cut-off value of the sFlt-1/ANG-II ratio was 3.06 showing a sensitivity (Se) of 96% and specificity (Sp) of 88.6% for severe pneumonia. The Se and Sp were 100% and 71.6% for ICU admission and 100% and 70.5% for intubation, respectively (Table 4).

### 3.5. Hypothetical Molecular Mechanisms Contributing to the Pathogenesis of Severe COVID-19 in Pregnant Women

In pregnant women with severe pneumonia by COVID-19, plasma levels of ANG-II are reduced, and plasma levels of sFlt-1 are increased, compared to those with non-severe disease. This leads to an imbalance in the sFlt-1/ANG-II ratio. Although the molecular mechanisms involved in the production of ANG-II and sFlt-1 were not explored in our work, current evidence allows us to propose a hypothetical pathway on how SARS-CoV-2 infection in the placenta affects the RAS signaling pathway contributing to the pathogenesis of severe COVID-19 in pregnant women (Figure 3). 

## 4. Discussion

### 4.1. Main Findings

A high ratio of sFlt-1/ANG-II was associated with a 1.31-fold increase in severe pneumonia and higher odds of ICU admission, viral sepsis, and maternal death. So, this ratio can be considered as a high-performance prognostic marker in pregnant women with COVID-19.

### 4.2. Comparison with Existing Literature

Studies in non-pregnant individuals have reported higher serum levels of sFlt-1 in patients with pneumonia due to COVID-19 compared to non-COVID-19-pneumonia [28]. Negro and colleagues have reported higher levels of sFlt-1 among deceased compared to COVID-19 survivors [29]. This report has demonstrated higher sFlt-1 levels in pregnant women with COVID-19 severe pneumonia. Other studies have reported lower plasma ACE-2 levels in non-survivors than in critically ill patients that have survived COVID-19 [30], however, our study has failed to demonstrate a difference in this parameter between those with critical illness and non-critical illness. We have found lower plasma levels of ANG-II in pregnant women with severe pneumonia by COVID-19. Lower ANG-II levels have been found in previous studies among people with acute respiratory distress syndrome not related to SARS-CoV-2. Studies in non-pregnant participants with severe COVID-19 have shown lower serum levels of ANG-II among deceased patients when compared to survivors [30]. A possible explanation of the ANG-II downregulation could be a defect in the endothelial–bound ACE activity due to endothelial injury [31].

We have shown that the sFlt-1/ANG-II ratio could be a potential predictor of adverse events such as severe pneumonia, ICU admission, intubation, viral sepsis, and death among pregnant women who tested positive for SARS-CoV-2 infection. This ratio should be tested in a larger cohort to prove its utility before its clinical use.

In relation to preeclampsia, no significant differences in the incidence among patients with severe and non-severe COVID-19 have been observed; this finding is contradictory to previous studies in which higher rates of preeclampsia have been demonstrated in cases of severe COVID-19 [32,33,34].

### 4.3. Strengths and Limitations

The strength of our study is the number of adverse events that allowed us to make statistical inferences for outcomes such as severe pneumonia. Furthermore, the baseline clinical characteristics between groups were similar, which decreases the probability of selection bias.

The limitations are, despite being a cohort, the analysis was carried out cross-sectionally, which does not allow us to infer the causal relationship between the sFlt-1/ANG-II ratio and adverse outcomes, and the ORs could be overestimated. Although the sFlt-1/ANG-II ratio has a positive predictive value for predicting the severity of COVID-19 in the short term, the results need clinical validation in a new cohort.

### 4.4. Clinical Interpretation

In this study, a negative correlation has been found between the plasma concentrations of ANG-II and sFlt-1. A high sFlt-1/ANG-II ratio is associated with several adverse outcomes related to COVID-19, such as severe pneumonia, ICU admission, intubation, viral sepsis, and death. The sFlt-1/ANG-II ratio may allow the development of predictive models for the identification of high-risk pregnant women in need of intensive surveillance and aggressive supportive treatment upon admission to the hospital, thus preventing clinical deterioration.

## 5. Conclusions

sFlt-1/ANG-II ratio is a promising predictor for adverse outcomes such as pneumonia, ICU admission, intubation, viral sepsis, and death in pregnant women with COVID-19. However, further research in a larger prospective cohort is needed to validate the association and accuracy of the sFlt-1/ANG-II ratio for the prediction of adverse events among pregnant women with COVID-19.

## Figures and Tables

**Figure 1 viruses-13-01906-f001:**
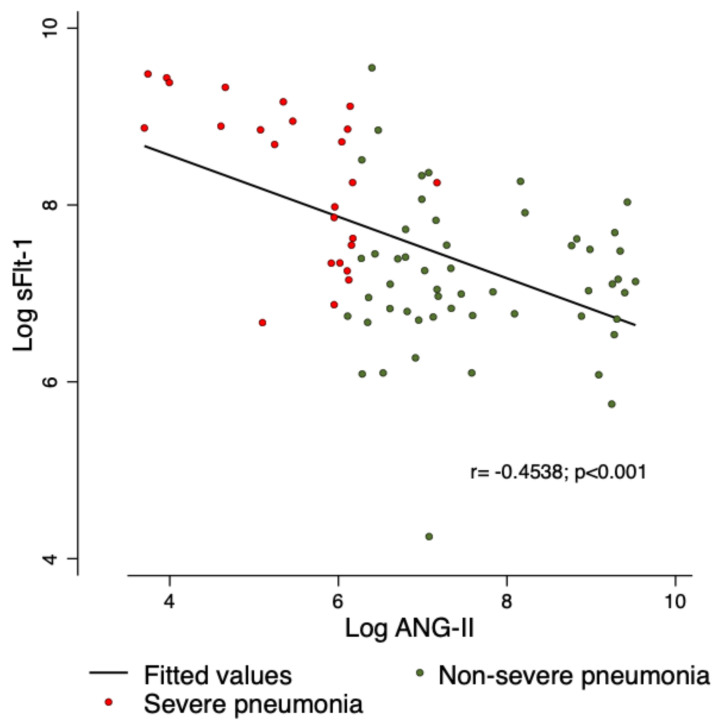
Relationship of sFlt-1 and ANG-II. A significant negative correlation was identified, between lower plasma concentrations of ANG-II, and very high plasma concentrations of sFlt-1.

**Figure 2 viruses-13-01906-f002:**
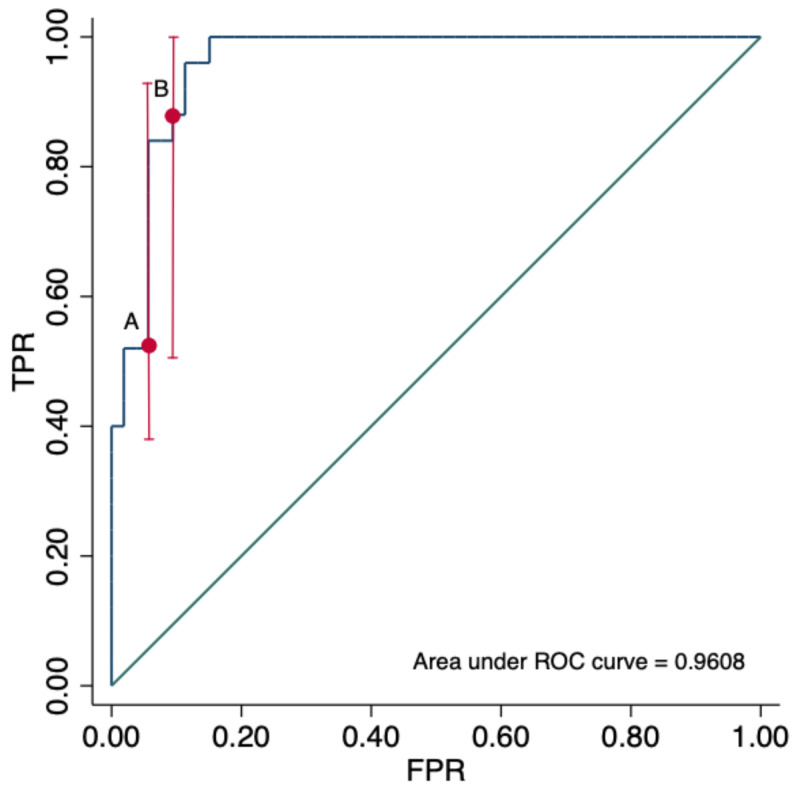
Area under the receiver-operating-curve (ROC) of sFlt1-1/ANG-II ratio for the prediction of severe pneumonia by COVID-19. ROC 0.9608. The detection rate (true-positive rate [TPR]) for severe pneumonia, at (A) 5% and (B) 10% false-positive rate (FPR) were 52% and 88%, respectively.

**Figure 3 viruses-13-01906-f003:**
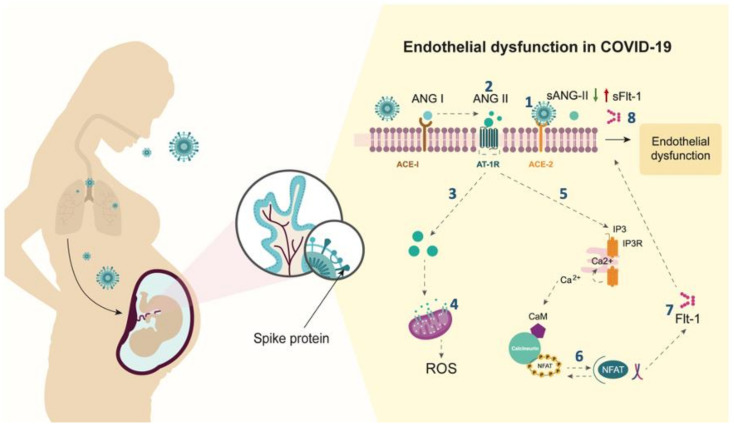
Hypothetical molecular mechanisms contributing to the pathogenesis of severe COVID-19 in pregnant women. Spike protein of SARS-CoV-2 virus binds to trophoblastic cells expressing ACE-2, (1) blocking the conversion of ANG-II into ANG 1-7 [3]. (2) Accumulation of ANG-II on the cell surface enhances its binding to the AT-1 receptor (AT-1R), promoting downstream signaling, followed by rapid endocytosis of the ANG-II/AT-1R complex [24]. (3) By avoiding the endosome/lysosome degradation, the excess of ANG-II is accumulated in endothelial cells [25]. (4) ANG-II binds to mitochondrial AT-1R [26], inducing cellular senescence with positive regulation of reactive oxygen species (ROS) [27]. 5) Over-activation of AT-1R on the cell membrane leads to increased PKC and calcineurin activity [25]. (6) Transcription factors NF-kB and NFAT are activated and translocated to the nucleus, leading to an increase in gene expression and release of Flt-1 [25]. (7) Flt-1 alternative splicing generates sFlt-1 isoform [25]. (8) The excess of sFlt-1 protein is released into the circulation causing endothelial dysfunction.

**Table 1 viruses-13-01906-t001:** Clinical characteristics of the study population.

Characteristic	Non-Severe COVID-19 *n* = 55	Severe COVID-19 *n* = 25	*p*-Value
Maternal age (years)	29.05 (24.94–33.5)	30.56 (28.40–33.73	0.185
Gestational age at diagnosis (weeks)	33.4 (28.0–38.1)	32.0 (27.2–36.1)	0.557
pBMI (kg/m^2^)	29.72 (25.0–33.8)	28.2 (23.4–33.5)	0.739
MAP (mmHg)	87.7 (82.7–95.0)	86.0 (80.0–89.7)	0.301
Smoking	1 (1.82%)	0	0.497
Chronic hypertension	3 (5.45%)	1 (4.00%)	0.782
Pre-gestational diabetes	3 (5.45%)	0	0.234
Asthma	1 (1.82%)	0	0.497
Chronic renal disease	4 (7.27%)	1 (4.00%)	0.575
SpO2%	94.5 (92.5–96.0)	92.5 (78–97.5)	0.713
Preeclampsia (clinical diagnosis)	11 (20.75%)	5 (20.0%)	0.939
True preeclampsia (Suspected preeclampsia + anormal sFlt-1/PlGF ratio)	6 (10.9%)	2 (8.0%)	0.118
Threatened preterm labor	2 (3.77%)	1 (4.00%)	0.961
Fetal growth restriction	4 (7.55%)	5 (20.0%)	0.108
Stillbirth	0	1 (4.00%)	0.143
Pneumonia	0	25 (100%)	<0.0001
ICU admission	0	11 (44.0%)	<0.0001
Intubation	0	7 (31.82%)	<0.0001
Viral sepsis	0	3 (12.0%)	0.009
Multiple organ dysfunction	0	3 (12.0%)	0.009
Maternal death	0	2 (8.00%)	0.034

pBMI: pregestational body mass index; MAP: Mean arterial pressure; SpO2: Oxygen saturation. Mann–Whitney-U test for continuous variables expressed as median and interquartile range; X^2^ or Fisher’s test for categorical variables expressed as number and percentage.

**Table 2 viruses-13-01906-t002:** Biochemical characteristics of the included population.

Characteristic	Non-Severe COVID-19 *n* = 55	Severe COVID-19 *n* = 25	*p*-Value
Leukocytes (×10/L)	8.15 (7.2–10.1)	8.5 (7.1–13.5)	0.339
Neutrophils (×10/L)	6.40 (5.30–7.60)	7.1 (5.6–12.6)	0.093
Lymphocytes (×10/L)	1.30 (1.0–1.5)	1.0 (0.6–1.4)	0.071
Hemoglobin (g/dL)	12.4 (11.3–13.9)	11.9 (11–12.7)	0.086
Hematocrit %	37.6 (34.0–41.6)	35.7 (32.6–38.7)	0.245
Platelets (×10^3^/L)	212 (184–270)	227 (170–271)	0.975
Glucose (mg/dL)	78.0 (73–85)	84 (72–120)	0.260
Creatinine (mg/dL)	0.55 (0.49–0.64)	0.54 (0.46–0.67)	0.624
Uric acid (mg/dL)	4.4 (3.8–5.8)	3.9 (3.4–5.0)	0.285
AST (U/L)	20.5 (17–28)	26 (21–36)	0.042
ALT (U/L)	17.5 (12–25)	23 (17–40)	0.082
LDH (U/L)	173 (146–212)	197 (152–295)	0.112
Direct bilirubin (mg/dL)	0.10 (0.06–0.14)	0.19 (0.07–0.42)	0.029
Indirect bilirubin(mg/dL)	0.32 (0.25–0.43)	0.34 (0.28–0.48)	0.464
Triglycerides (mg/dL)	263 (203–313)	265 (210–312)	0.885
Total cholesterol (mg/dL)	197 (172–235)	154 (118–217)	0.017
D-dimer (ng/mL)	1549 (1242–2981)	1438 (1248–2511)	0.302
Fibrinogen (mg/dL)	526 (481–591)	570 (428–611)	0.521
PTT (seconds)	26.2 (24.8–29.2)	26.9 (24.8–28.9)	0.949
PT (seconds)	10.8 (10.55–11.4)	10.3 (9.9–11)	0.398
C-RP (mg/L)	21.1 (6.45–81.7)	61.15 (16.5–188)	0.014
Procalcitonin (ng/mL)	0.05 (0.03–0.13)	0.2 (0.07–0.53)	0.0006
PlGF (pg/mL)	150.1 (56–215.6)	114.3 (32.29–212.3)	0.186
sFlt-1 (pg/mL)	1424 (1054–2099)	6119 (2099–7900)	0.0001
ACE-2 (pg/mL)	8754 (6040–27480)	7904 (5928–14216)	0.324
ANG-II (pg/mL)	1479 (915.3–7873)	404.3 (180.8–471)	0.0001
sFlt1/PlGF ratio	11.21 (5.43–26.38)	53.72 (31.87–126.12)	0.0001
sFlt-1/ANG-II ratio	0.92 (0.25–2.03)	14.27 (4.47–42.46)	0.0001

AST: Aspartate aminotransferase; ALT: Alanine aminotransferase; LDH: Lactate dehydrogenase; PTT: Partial thromboplastin time; PT: prothrombin time; C-RP: C-reactive protein; PlGF: Placental growth factor; sFlt-1: Soluble fms-like tyrosine kinase-1; ACE-2: Angiotensin-converting enzyme-2; ANG-II: Angiotensin-II. Mann-Whitney-U test for continuous variables expressed as median and interquartile range.

**Table 3 viruses-13-01906-t003:** Association between biochemical markers and severe COVID-19.

Biochemical Marker	OR	95% CI	*p*-Value
AST (U/L)	1.00	0.99–1.00	0.636
Direct bilirubin (mg/dL)	15.69	0.81–303.44	0.069
Total cholesterol (mg/dL)	0.99	0.98–1.00	0.064
C-RP (mg/L)	1.01	1.00–1.02	0.025
Procalcitonin (ng/mL)	1.12	0.67–1.88	0.651
sFlt1 (pg/mL)	1.01	1.00–1.01	<0.0001
ANG-II (pg/mL)	0.99	0.98–0.99	0.001
sFlt1/PlGF	1.02	1.00–1.03	0.002
sFlt-1/ANG-II	1.31	1.09–1.56	0.003

C-RP: C-reactive protein; sFlt-1: Soluble fms-like tyrosine kinase-1; ANG-II: Angiotensin-II; PlGF: Placental growth factor; OR: Odds ratio; CI: Confidence interval.

**Table 4 viruses-13-01906-t004:** Performance of sFlt-1/ANG-II ratio ≥ 3.06 for the prediction of adverse maternal outcomes.

Outcome	Se (95% CI)	Sp 95% CI	Positive LR 95% CI	Negative LR (95% CI)
Severe pneumonia	0.96 (0.88–1.0)	0.886 (0.80–0.972)	8.48 (3.97–18)	0.045 (0.01–0.31)
ICU admission	1.0 (1.0–1.0)	0.716 (0.443–0.789)	3.52 (2.26–4.95)	0.01 (0.01–0.88)
Intubation	1.0 (1.0–1.0)	0.705 (0.567–0.784)	3.4 (1.93–4.20)	0.01 (0.01–1.54)
Viral sepsis	1.0 (1.0–1.0)	0.64 (0.531–0.748)	2.77 (1.50–3.89)	0.01 (0.01–2.63)
Maternal death	1.0 (1.0–1.0)	0.631 (0.523–0.74)	2.71 (1.26–4.04)	0.01 (0.01–3.34)

sFlt-1: Soluble fms-like tyrosine kinase-1; ANG-II: Angiotensin-II; ICU: Intensive care unit; Se: sensitivity; Sp: specificity; LR: Likelihood ratio; CI: Confidence interval.

## Data Availability

The data presented in this study are available on request from the corresponding author. The data are not publicly available due to privacy.

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
