# Peer review of "Novel Ratio Soluble Fms-like Tyrosine Kinase-1/Angiotensin-II (sFlt-1/ANG-II) in Pregnant Women Is Associated with Critical Illness in COVID-19"

_viruses, 2021, doi:10.3390/v13101906_

Round 1

Reviewer 1 Report

This is a well written manuscript describing the sFlt-1/ANG-II ration as a marker for severe  COVID-19 in pregnancy women. I have only a couple of comments/suggestions to make. The main one is that the authors state on lines 144-145 that the sFlt-1/PIGF ration does not differ between study groups yet in Table 2  there is a significant difference reported. The authors should consider citing DOI:10.1111/1471-0528.16339 in this regard.

I would suggest the authors make some comments about the pregnancy outcome in their groups. Did the severe COVID-19 group also have fetal/infant placental morbidity/mortality?

Reviewer 2 Report

The manuscript titled "Novel ratio soluble fms-like tyrosine kinase-1/Angiotensin-II (sFlt-1/ANG-II) in pregnant women is associated with critical illness in COVID-19" aims to examine the relationship between an angiogenic marker (sFlt-1) and RAS component (ANG-II) with adverse outcomes in pregnant women with COVID-19. The introduction provides a well-written context and content for the premise of this study. This manuscript identifies the need to examine the sFlt/ANG-II ratio in pregnant women with COVID-19 as it could be a critical/ promising predictor for adverse events. The objective can is stated in the last paragraph of the introduction in this manuscript. The objective is clear and concise. The methods of this study are appropriate for the study. Standard medical record patient information and blood samples for pregnant women were obtained, and descriptions and biochemical characteristics of the cohort were presented in Tables 1 and 2.
The authors describe the primary (severe pneumonia) and secondary outcomes (ICU admission, intubation, viral sepsis, and maternal death) of pregnant women with COVID-19, and they use standard criteria to substantiate each outcome. The author explains how differences between variables with COVID-19 severity were compared using the Mann-Whitney U test or X2 test for statistical analysis. The participant size in this study is large, which allows for more accurate values and subsequent conclusions to be drawn. The author described a significant association between severe pneumonia in pregnant women with COVID-19 infection and a high sFlt-1/ANG-II ratio. By recording primary and secondary outcomes and baseline characteristics, the authors concluded that there are lower plasma levels of ANG-II and higher sFlt-1 in pregnant women in the severe COVID-19 group than in the non-severe group. The author's conclusions regarding this current study are appropriate with the relevant data presented. The author also offers a hypothetical pathway that may contribute to the pathogenesis of severe COVID-19 in pregnant women, which could pave the way for future work. Overall, this manuscript is well-written with minimal to no grammatical/ spelling errors. The reviewers believe that the data presented in this study uses appropriate tables, graphs, and pathways to convey the findings of this study to the readers.

Author Response

Thank you for your comment.